

# Improving physical activity, sedentary behaviour and sleep in COPD: perspectives of people with COPD and experts via a Delphi approach

Hayley Lewthwaite[1], Tanja W. Effing[2,3], Anke Lenferink[4,5], Tim Olds[1] and Marie T. Williams[1]

[1] Alliance for Research in Exercise, Nutrition and Activity, School of Health Sciences, University of South Australia, Adelaide, South Australia, Australia
[2] College of Medicine & Public Health, Flinders University of South Australia, Bedford Park, South Australia, Australia
[3] Department of Respiratory Medicine, Southern Adelaide Local Health Network, Bedford Park, South Australia, Australia
[4] Department of Pulmonary Medicine, Medisch Spectrum Twente, Enschede, Netherlands
[5] Department of Health Technology and Services Research, Faculty of Behavioural Sciences, University of Twente, Enschede, Netherlands

Corresponding author
Hayley Lewthwaite,
Hayley.Lewthwaite@mymail.
unisa.edu.au

## ABSTRACT

**Background.** Little is known about how to achieve enduring improvements in physical activity (PA), sedentary behaviour (SB) and sleep for people with chronic obstructive pulmonary disease (COPD). This study aimed to: (1) identify what people with COPD from South Australia and the Netherlands, and experts from COPD- and non-COPD-specific backgrounds considered important to improve behaviours; and (2) identify areas of dissonance between these different participant groups.

**Methods.** A four-round Delphi study was conducted, analysed separately for each group. Free-text responses (Round 1) were collated into items within themes and rated for importance on a 9-point Likert scale (Rounds 2–3). Items meeting *a priori* criteria from each group were retained for rating by all groups in Round 4. Items and themes achieving a median Likert score of $\geq 7$ and an interquartile range of $\leq 2$ across all groups at Round 4 were judged important. Analysis of variance with Tukey's post-hoc tested for statistical differences between groups for importance ratings.

**Results.** Seventy-three participants consented to participate in this study, of which 62 (85%) completed Round 4. In Round 4, 81 items (PA $n = 54$; SB $n = 24$; sleep $n = 3$) and 18 themes (PA $n = 9$; SB $n = 7$; sleep $n = 2$) were considered important across all groups concerning: (1) symptom/disease management, (2) targeting behavioural factors, and (3) less commonly, adapting the social/physical environments. There were few areas of dissonance between groups.

**Conclusion.** Our Delphi participants considered a multifactorial approach to be important to improve PA, SB and sleep. Recognising and addressing factors considered important to recipients and providers of health care may provide a basis for developing behaviour-specific interventions leading to long-term behaviour change in people with COPD.

# INTRODUCTION

For both the general population (*Buman et al., 2014*; *Loprinzi, Lee & Cardinal, 2014*; *Biswas et al., 2015*; *Chastin et al., 2015*) and people with chronic obstructive pulmonary disease (COPD) (*Nunes et al., 2009*; *Omachi et al., 2012*; *Gimeno-Santos et al., 2014*; *Geiger-Brown et al., 2015*; *Furlanetto et al., 2017*), being more physically active, spending less time in prolonged sedentary behaviour and getting adequate sleep reduces the risk of hospitalisation and death, improves quality of life and alleviates symptoms. Despite this, people with COPD, like many in the general population, spend a large part of the day sitting or lying rather than in more active pursuits and have poor sleep quantity and quality (*Hunt et al., 2014*).

Interventions aiming to change behaviour in people with COPD have previously been developed by healthcare providers and/or researchers, with little or no input from the intended participants. The lack of consistent, long-term positive effects of many behaviour change interventions in this population (*Soler, Diaz-Piedra & Ries, 2013*; *McDonnell et al., 2014*; *Lahham, McDonald & Holland, 2016*; *Mantoani et al., 2016*; *Mesquita et al., 2017a*; *Mesquita et al., 2017b*; *Williams et al., 2017*) may in part be due to intervention designers and intervention participants having different ideas about what is important. This notion has recently been demonstrated where people with COPD and experts managing COPD were asked about different aspects of the disease (*Celli et al., 2017*). Experts often perceived symptoms such as fatigue and tiredness to have a lesser impact on quality of life for people with COPD, and for daily activities to be less affected by the disease (*Celli et al., 2017*). Recognising and addressing factors important to both people with COPD and healthcare professionals managing COPD may provide a better basis for developing interventions to facilitate behaviour change.

With the recent shift toward exploring strategies to improve sedentary behaviour and sleep in people with COPD, obtaining the perspectives of experts from non-COPD-specific backgrounds provides an opportunity to identify potential novel strategies to improve these behaviours. Furthermore, it is possible that due to differences in social, cultural, environmental and policy factors, people with COPD from different geographical locations may differ in their perspectives about what is important to change behaviours. In this study, we had an opportunity to include people with COPD from Australia and the Netherlands, two countries where differences in factors likely to influence behaviours exist (e.g., greater provision of funding and policies in the Netherlands to support safe cycling) (*ABS, 2006*; *Cloïn, 2012*). Two authors (TE, AL) maintain professional connections in the Netherlands and are bilingual in Dutch and English.

The primary aim of this study was to identify factors considered as important to people with COPD from different geographical locations and experts from COPD- and non-COPD-specific backgrounds to improve physical activity (PA), sedentary behaviour

(SB) and sleep. The secondary aim was to identify areas of dissonance between factors important to these different participant groups.

## MATERIAL AND METHODS

### Design

This research study followed a Delphi design, reported following recommendations by *Diamond et al. (2014)* and *Sinha, Smyth & Williamson (2011)* for the Delphi process. Ethical approval was granted by The Human Research Ethics Committees of Southern Adelaide Local Health Network (#516.15), Medisch Spectrum Twente (#K16-09) and University of South Australia (#0000034584).

### Participants

Four participant groups were purposefully recruited: people with COPD from Adelaide, South Australia (SA-COPD) or Enschede, the Netherlands (NL-COPD), and international experts in the clinical management of COPD (COPD-E) or non-COPD-specific public health strategies (Non-COPD-E).

Participants for COPD groups were identified from pulmonary rehabilitation and physiotherapy databases of the Repatriation General Hospital, Adelaide (SA-COPD) or Medisch Spectrum Twente, Enschede, the Netherlands (NL-COPD). For inclusion in this study, people with COPD needed to have at least mild COPD confirmed by spirometry (http://goldcopd.org/) and have participated in a pulmonary rehabilitation program or ongoing supervised physiotherapy exercise program within the 12 months prior to recruitment.

For inclusion in this study, experts in the clinical management of COPD needed to be an author of one of two clinical practice guidelines (*Yang et al., 2017*; *Qaseem et al., 2011*) or relevant position statements (*Watz et al., 2014*; *Spruit et al., 2013*). Experts in non-COPD-specific public health strategies were eligible if they were an author of a health promotion initiative, healthy lifestyle program or public health policy published by an authoritative body in Australia or anywhere else in the world. Relevant documents were identified by screening websites of national and international government health departments or by searching the internet (e.g., with search terms: public health policy, health promotion initiative). With an extensive list of potential participants identified, one of our research team with expertise in public health (TO) screened the list and provided recommendations around prioritisation of study invitations.

Eligible participants were sent a study information pack via post (SA-COPD only) or email. Participants who responded to the invitation and provided written consent were enrolled into the study. Where potential participants for the SA-COPD group did not contact the research team following initial invitation, they were followed up by phone. We aimed to recruit a minimum of 15 participants for each group. While there is no consensus as to the optimal number of participants for a Delphi study, historically, Delphi studies have comprised 11–25 participants (*Diamond et al., 2014*), with concerns that fewer participants might result in underrepresentation of opinion (*Hsu & Sandford, 2007*).

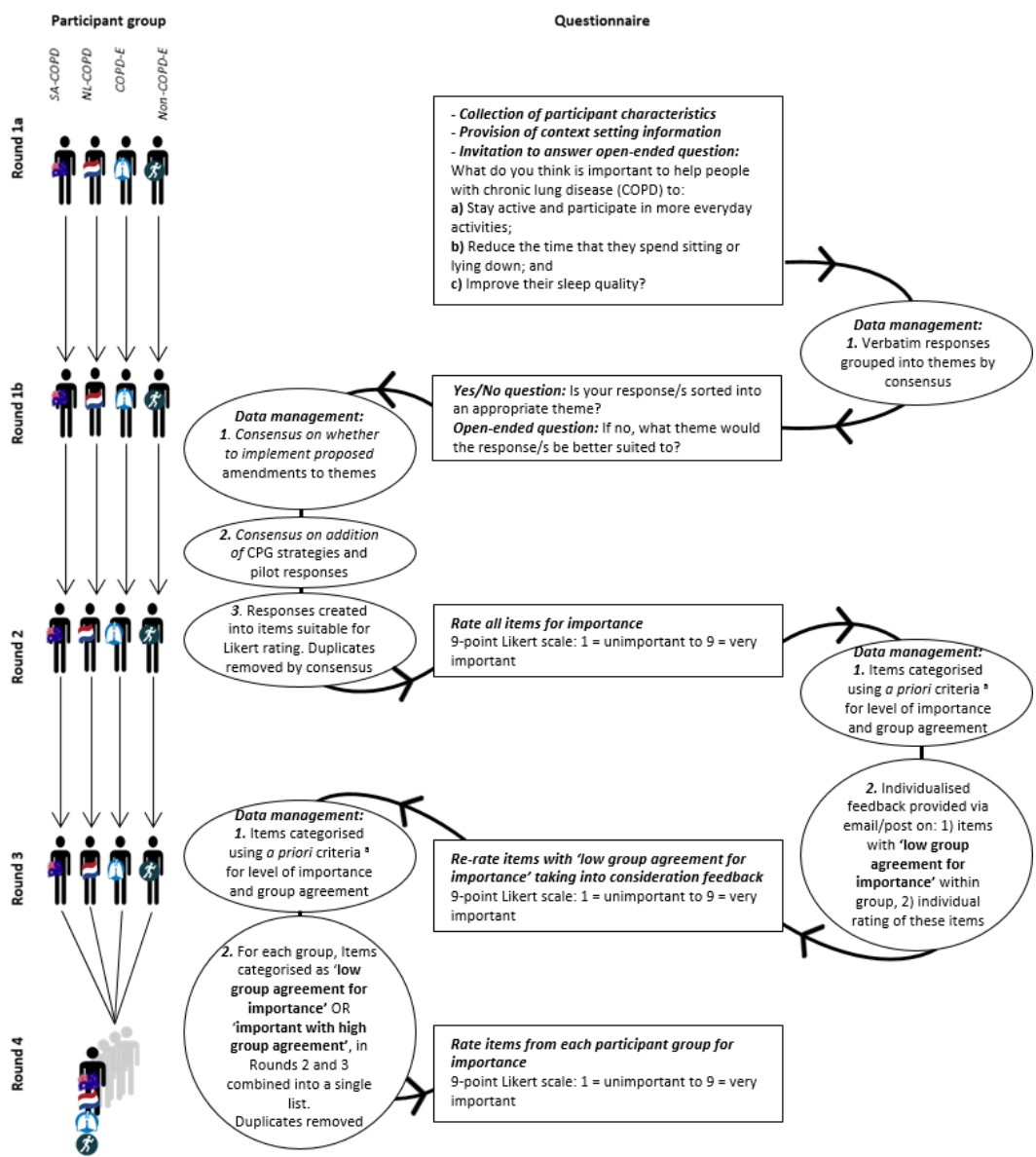

**Figure 1  Delphi procedure.** CPG, clinical practice guideline; [a], Items categorised according to group me-
dian Likert score and interquartile range (IQR): (1) low group agreement for importance = IQR > 2; (2)
important with high group agreement = Median ≥ 7 and IQR ≤ 2; or (3) unimportant with high group
agreement = Median < 7 and IQR ≤ 2.

## Procedure

Four Delphi questionnaire rounds were prospectively planned, conducted electronically
via Survey Monkey® or as hard copy via post (SA-COPD only). Figure 1 outlines the
Delphi procedure. Questionnaires were group-specific for Rounds 1–3. For Round 4, all
participant groups completed the same questionnaire. All questionnaire rounds for the
NL-COPD group were conducted in Dutch, translated by two authors independently (TE,
AL). Rounds 1–4 were analysed separately for each participant group.

Participant anonymity was maintained throughout the study by use of individualised participant communication electronically or via post. Participants were given three weeks to complete each round. A reminder was sent at one and two weeks following initial invitation. Where participants missed a round, they were invited to participate in proceeding rounds unless they requested to withdraw.

The initial draft questionnaire for Round 1 was pilot tested by participants representative of each participant group. The purpose of the pilot was to clarify the expected time taken to complete the questionnaire, optimisation of questionnaire structure and clarity of wording. Following minor refinements based on feedback (Table S1), the Round 1 questionnaire was finalised.

## Delphi questionnaires

For the **Round 1** questionnaire, participants were provided context-setting information (Fig. 2) and invited to answer the open-ended question (Round 1a): 'What do you think is important to help people with chronic lung disease (COPD) to:

a) Stay active and participate in more everyday activities;
b) Reduce the time they spend sitting and lying down; and
c) Improve their sleep quality?'

Free-text responses were collated into common themes by two authors independently (HL & TO; HL & MTW; TE & AL). Individual participants were sent a copy of their responses and the theme into which they had been allocated (Round 1b), providing participants with an opportunity to confirm whether their response/s had been sorted into an appropriate theme.

The list of Round 1a responses was compared to strategies derived from a recent systematic review of COPD management guidelines (*Lewthwaite et al., 2017*), and responses from the pilot of the Round 1 Delphi questionnaire. Items within the pilot and systematic review not already represented by Round 1a participant responses were added to the list for each participant group. This ensured a comprehensive list of items was available for rating in subsequent rounds. All responses were created into items suitable for Likert scale rating. Duplicate items were removed by consensus among two authors (HL, MTW).

For **Round 2–3** questionnaires, each group was sent their corresponding list of items collated into common themes, and invited to rate each item for importance on a 9-point Likert scale (1 = unimportant to 9 = very important). At the completion of Round 2 and 3, items were categorised using *a priori* criteria based on the group median Likert score and interquartile range (IQR) where:

1) low group agreement for level of importance was defined as an IQR > 2 (item continued to next round for rating);
2) important with high group agreement was defined as a median ≥7 and IQR ≤ 2 (item retained for rating in Round 4); and
3) unimportant with high group agreement was defined as a median <7 and IQR ≤ 2 (item removed from subsequent rounds).

For the **Round 3** questionnaire only, each participant was sent an individualised report of Round 2 results appropriate to their group via email or post (SA-COPD only). Controlled

Many people with chronic lung diseases have trouble with physical

activities, such as planned exercise, walking or cycling and

everyday activities such as household chores, shopping and

gardening. People with chronic lung disease therefore spend a lot of

time throughout the day sitting or lying down and have difficulties

sleeping at night. While pulmonary rehabilitation and supervised

exercise programs may improve fitness, it may not change the way

people with chronic lung disease spend their time in everyday life.

We do not currently know how we can help people with chronic

lung disease to stay active once the pulmonary rehabilitation

program is completed.

**Figure 2 Context setting information.**

feedback was provided on: (1) items that met criteria as 'low group agreement for level of importance'; and (2) their individual rating of these items. This provided an opportunity for participants to reconsider their rating of items with low group agreement from Round 2, in order to encourage more decisive ratings toward unimportance or importance in Round 3.

For the **Round 4** questionnaire, all participants were provided with, and invited to rate, the same list of items collated into common themes. This list comprised all items retained from Round 2 and Round 3 within each participant group that met criteria for 'low group agreement for level of importance' or 'important with high group agreement'. Duplicate items were removed by consensus among two authors (HL, MTW). The finalised list was translated from English to Dutch independently by two authors (TE, AL) for the NL-COPD participants.

## Data analysis

Participant groups were characterised using descriptive statistics with findings tabulated for comparison across groups. To identify items and themes considered as important across all participant groups *(primary aim)*, the median Likert score and IQR of all items at the end of Round 4 was calculated for each group. Items and themes achieving a median of $\geq 7$ were considered as important (*Fitch et al., 2001*). Based on previous Delphi studies (*Jones & Hunter, 1995*; *Rayens & Hahn, 2000*; *Vandelanotte et al., 2010*; *Effing et al., 2016*), high
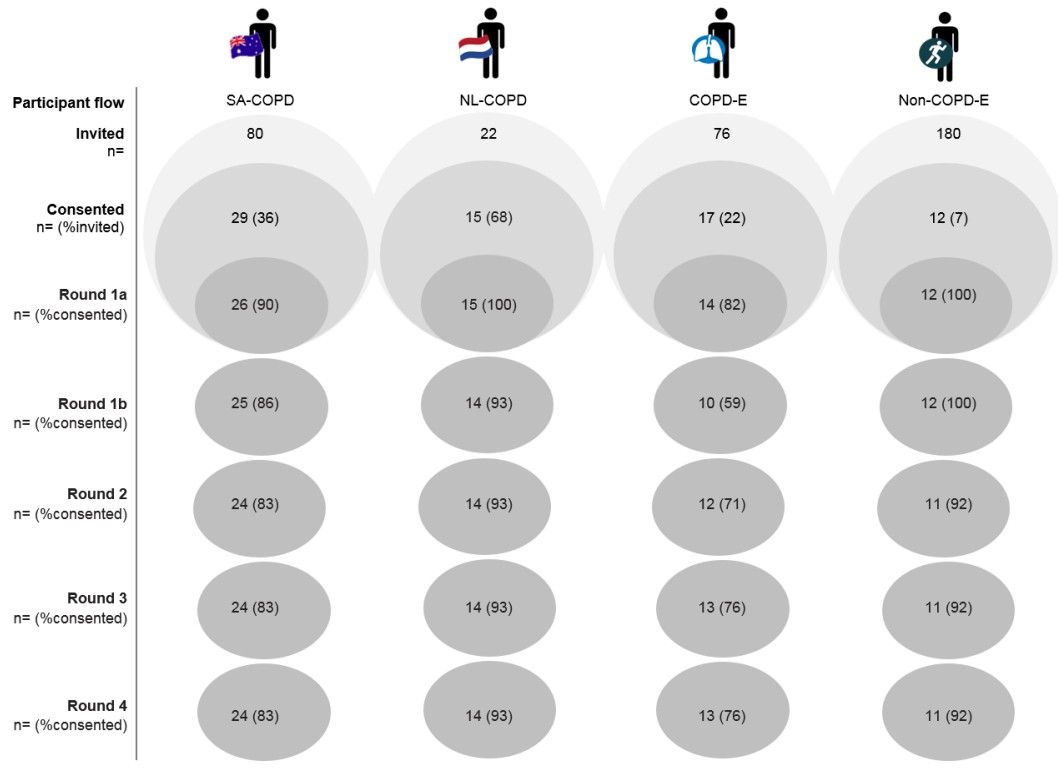

**Figure 3** Participant flow.

group agreement was defined as an IQR was ≤2. Items and themes that did and did not meet criteria for 'important with high group agreement' across all participant groups at the end of Round 4 were summarised and reported. To enable group comparisons *(secondary aim)*, items and themes that did and did not meet criteria for 'important with high group agreement' for each of the four participant groups were tabulated. Analysis of variance with Tukey's post-hoc tested for statistical differences between groups for mean rating of items within each theme. Sequential Bonferroni corrections were applied to reduce risk of type one error with multiple comparisons.

## RESULTS

Seventy-three participants were included in this study. Figure 3 outlines the participant flow from Rounds 1 to 4. Participant groups are described in Table 1.

### Responses, Round 1–3 questionnaires

In Round 1a, there were a total of 408 participant responses to the open-ended question sorted into common themes for PA (SA-COPD $n = 13$; NL-COPD $n = 7$; COPD-E $n = 11$; Non-COPD-E $n = 9$), SB (SA-COPD $n = 11$; NL-COPD $n = 7$; COPD-E $n = 11$; Non-COPD-E $n = 7$) and sleep (SA-COPD $n = 7$; NL-COPD $n = 5$; COPD-E $n = 7$; Non-CO PD-E $n = 8$). High agreement was reported by participants' in Round 1b for allocation of their individual responses to themes, with few amendments required (Table S2).

**Table 1  Baseline characteristics of participant groups.**

|  | COPD groups | | Expert groups | |
| --- | --- | --- | --- | --- |
|  | SA-COPD | NL-COPD | COPD-E | Non-COPD-E |
| Completed Round 1a ($n =$) | 26 | 15 | 14 | 12 |
| Age (years) | 66.8 (±9.7) | 69.9 (±6.1) | 50.1 (±11.5) | 43.5 (±11.7) |
| Gender (%male) | 54 | 60 | 43 | 50 |
| Education ($n$, %) |  |  |  |  |
|    Left school before year 12 | 13 (50) | 2 (13) | – | – |
|    Completed year 12 | 4 (15) | 8 (53) | – | – |
|    Technical degree or diploma | 6 (23) | 5 (33) | – | – |
|    University degree(s) | 3 (12) | 0 (0) | – | – |
| Profession ($n$, %) | – | – |  |  |
|    Academic/research | – | – | 6 (43) | 10 (83) |
|    Physician | – | – | 4 (29) | 2 (17) |
|    Nurse | – | – | 2 (14) | 0 (0) |
|    Allied health professional | – | – | 2 (14) | 0 (0) |
| Number of self-reported medical conditions[*] | 4.1 (±2.4) | 2.3 (±1.2) | – | – |
|    COPD ($n$, %) | 23 (88) | 15 (100) |  |  |
|    Emphysema ($n$, %) | 13 (50) | 1 (7) |  |  |
|    Bronchitis ($n$, %) | 5 (19) | 3 (20) |  |  |
|    Asthma ($n$, %) | 11 (42) | 3 (20) |  |  |
|    Self-reported sleep disorder ($n =$ yes) | 4 (15) | 2 (13) |  |  |
|    Arrhythmia ($n$, %) | 2 (8) | 3 (20) |  |  |
|    High blood pressure ($n$, %) | 9 (35) | 3 (20) |  |  |
|    Depression ($n$, %) | 8 (31) | 0 (0%) |  |  |
|    Osteoporosis ($n$, %) | 9 (35) | 2 (13) |  |  |
| Smoking history ($n$, %) |  |  | – | – |
|    Never | 4 (15) | 0 (0) | – | – |
|    Former | 18 (69) | 15 (100) | – | – |
|    Current | 4 (15) | 0 (0) | – | – |
| Use supplemental oxygen ($n$, %yes) | 1 (4) | 2 (13) | – | – |
| Hospitalisation previous 12 months ($n$, %yes) | 4 (15) | 4 (27) | – | – |
|    Number of hospitalisations (median, range) | 1 (1–6) | 1 (1–2) |  |  |
| mMRC ($n$, %) |  |  | – | – |
|    0 | 4 (15) | 5 (33) | – | – |
|    1 | 15 (58) | 1 (7) | – | – |
|    2 | 2 (8) | 5 (33) | – | – |
|    3 | 4 (15) | 1 (7) | – | – |
|    4 | 1 (4) | 3 (20) | – | – |
| Self-report days active in previous week (median, IQR) | 1.5 (0–3) | 2.0 (2–4.5) | – | – |

**Notes.**

[*]$p \leq 0.05$ between group difference.

Results expressed as mean (±SD) unless otherwise stated.

mMRC, modified Medical Research Council dyspnoea score.

**Table 2** Summary of items important to all participant groups to improve physical activity, sedentary behaviour and sleep.

| Theme | Important items PA | Important items SB | Important items sleep |
|---|---|---|---|
| Non-specific | Be: Resilient; Compliant with medical care/medication; Accepting | – | – |
| Understand patient concerns/fears/expectations | Have self-motivation; Manage breathlessness/associated fear and expectations; Manage expectations around pain; Overcome fear of outdoors/infections | Manage breathlessness/associated fear; Motivation | – |
| Behaviour change/self-efficacy/autonomy | Build confidence/skills to cope/change behaviour; Feel in control/empowered; Self-regulate activity; Persevere; Seek education; Create helpful thoughts; Follow action plan | Create positive thoughts; Have self-confidence with managing symptoms and behaviour change | – |
| Manage symptoms | Manage: Breathlessness; Fatigue. Provide appropriate drug treatment. | Manage: Breathlessness; Fatigue | – |
| Self-monitoring/goal setting | Self-monitoring/goal setting to increase LPA and daily activities; Positive feedback | Plan and create goals for daily activities | – |
| Education | Education on: AECOPD; How to be active/what it means; Breathing exercises; Benefits LPA; Manage stress/anxiety | Education on: Benefit of daily activities/LPA; Consequences of being inactive; How to structure life to reduce SB; Mental health | – |
| Professional support | Provision of: Physician/advice encouragement/support; PR and maintenance programs; Exercise advice; F/U with multi-D support and assessment of PA; Self-management; Counselling/motivational interviewing | Provision of: Physician/advice encouragement/support; F/U AECOPD; Exercise plan; Supervised exercise program | – |
| Social support/interactions | Have supportive loved ones involved in care; Take part in social activities | Social interactions with friends/family | – |
| Accessible/affordable exercise facilities | Free exercise programs; Community gym close to home | – | – |
| Increase physical activity/fitness | Engage in: Daily activities; Regular exercise; Improve fitness. | Improve fitness | – |
| Manage co-existing problems/conditions | Manage: Pain; Cardiac conditions; Musculoskeletal conditions | Manage: Pain; Cardiac conditions; Musculoskeletal conditions | – |
| Modify/understand physical environment | Facilitative built environment, well-surfaced footpaths | – | – |
| Enjoyment | – | Social life; Hobbies; Scheduled activities | – |

**Table 2** (*continued*)

| Theme | Important items PA | Important items SB | Important items sleep |
|---|---|---|---|
| Modify/understand employment commitments | – | – | – |
| Increase/maintain daily activities | – | Be independent; Engage: ADL's/daily activities including outdoor and indoor chores; LPA; Exercise | – |
| Understand cause/treat sleep problem | – | – | – |
| Manage anxiety/stress/worry | – | – | Manage anxiety/stress/worry/intrusive thoughts; Relax |
| Follow sleep hygiene principles | – | – | Know and follow sleep hygiene principles |

**Notes.**

–, no item within theme met criteria for 'important with high group agreement' across all participant groups; AECOPD, acute exacerbation of chronic obstructive pulmonary disease symptoms; F/U, follow-up; multi-D, multidisciplinary; LPA, light-intensity physical activity; PA, physical activity; PR, pulmonary rehabilitation; SB, sedentary behaviour.

In Rounds 2–3, few items met criteria for 'unimportant with high group agreement' requiring removal from rating in subsequent rounds. Items considered as 'unimportant with high group agreement' in Rounds 2 or 3 for each participant group are presented in Table S3.

## Primary aim—items and themes important across all groups, Round 4

Of the 157 items rated in Round 4, 54 (65%) were considered as important by all participant groups to improve PA, 24 (47%) to improve SB and three (13%) to improve sleep. Table 2 provides a summary of important items organised into common themes.

For improving PA, all participant groups considered nine of the 14 themes (64%) important. These nine themes were: 'Manage symptoms', 'Understand patients' concerns/fears/expectations', 'Behaviour change/self-efficacy/autonomy', 'Self-monitoring/goal setting', 'Education', 'Professional support', 'Social support/interactions', 'Accessible/affordable exercise facilities' and 'Non-specific'. For improving SB, all participant groups considered seven of the 14 (50%) themes important: 'Manage symptoms', 'Understand patient concerns/fears/expectations', 'Behaviour change/self-efficacy/autonomy', 'Self-monitoring/goal setting', 'Education', 'Professional support' and 'Manage co-existing problems/conditions'. For improving sleep, all participant groups considered two of the seven (29%) themes important: 'Manage symptoms' and 'Follow sleep hygiene principles'. Figure 4 presents themes that did (indicated by bold text) and did not (indicated by greyed text) meet criteria for 'important with high group agreement' across all four participant groups.

## Secondary aim—differences between groups, Round 4

Table 3 presents themes that did and did not meet criteria for 'important with high group agreement' at the completion of Round 4 for each of the four participant groups. For average rating of items within themes, significant differences existed predominantly between

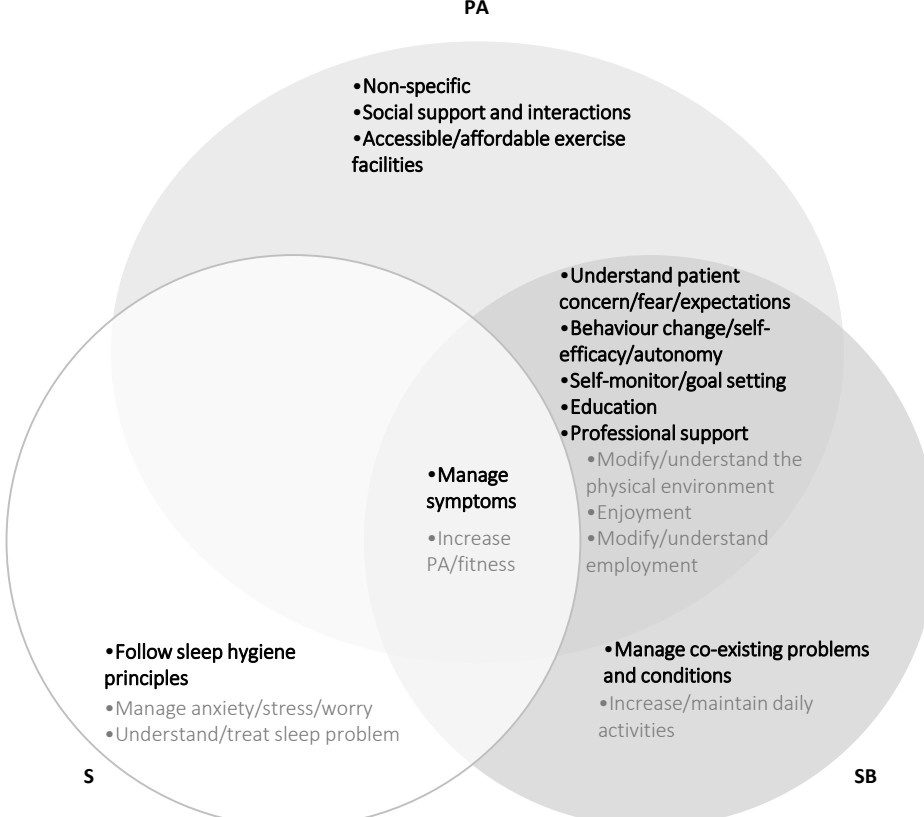

**Figure 4** **Themes that did and did not meet criteria for important with high group agreement across all four participant groups.** PA, physical activity; SB, sedentary behaviour; S, Sleep. Themes in bold text met criteria for important with high group agreement across all participant groups. Themes in greyed text met criteria for important with high group agreement across some but not all participant groups.

the NL-COPD group and remaining groups. The NL-COPD group rated items within 'Modify/understand employment commitments' significantly higher than all other groups to improve PA ($p = 0.001$) and SB ($p \leq 0.001$); and 'Accessible/affordable exercise facilities' higher than expert groups to improve PA ($p = 0.002$). Further differences existed where the Non-COPD-expert group rated items within 'Understand patients' concerns/fears and expectations' lower than the SA-COPD group to improve SB ($p = 0.002$) and the COPD-E group rated items within 'Increase physical activity/fitness' lower than the SA-COPD group to improve sleep ($p = 0.001$).

## DISCUSSION

This Delphi study aimed to identify factors to improve PA, SB and sleep considered as important to people living with COPD and experts with and without COPD specific backgrounds. The three key findings of this study were: **(1)** across all three behaviours, there was clear consensus for the importance of symptom/disease management and targeting behavioural factors, with a lesser focus on adapting the social or physical environments;

Lewthwaite et al. (2018), *PeerJ*, DOI 10.7717/peerj.4604

**Table 3**  Themes important with high group agreement for each participant group to improve physical activity, sedentary behaviour, and sleep.

| Theme | Physical activity | | | | Sedentary behaviour | | | | Sleep | | | |
|---|---|---|---|---|---|---|---|---|---|---|---|---|
| | COPD groups | | Expert groups | | COPD groups | | Expert groups | | COPD groups | | Expert groups | |
| | SA-COPD | NL-COPD | COPD-E | Non COPD-E | SA-COPD | NL-COPD | COPD-E | Non COPD-E | SA-COPD | NL-COPD | COPD-E | Non COPD-E |
| Non-specific[a] | ✓ | ✓ | ✓ | ✓ | ✓ | ✓ | X | X | ✓ | ✓ | X | ✓ |
| Understand patient concerns/fears/expectations | ✓ | ✓ | ✓ | ✓ | ✓ | ✓ | ✓ | ✓ | – | – | – | – |
| Behaviour change/self-efficacy/autonomy | ✓ | ✓ | ✓ | ✓ | ✓ | ✓ | ✓ | ✓ | – | – | – | – |
| Manage symptoms | ✓ | ✓ | ✓ | ✓ | ✓ | ✓ | ✓ | ✓ | ✓ | ✓ | ✓ | ✓ |
| Self-monitoring/goal setting | ✓ | ✓ | ✓ | ✓ | ✓ | ✓ | ✓ | ✓ | ✓ | ✓ | X | ✓ |
| Education | ✓ | ✓ | ✓ | ✓ | ✓ | ✓ | ✓ | ✓ | – | – | – | – |
| Professional support | ✓ | ✓ | ✓ | ✓ | ✓ | ✓ | ✓ | ✓ | – | – | – | – |
| Social support/interactions | ✓ | ✓ | ✓ | ✓ | X | ✓ | X | ✓ | – | – | – | – |
| Accessible/affordable exercise facilities | ✓ | ✓ | ✓ | ✓ | – | – | – | – | – | – | – | – |
| Increase physical activity/fitness | ✓ | ✓ | X | ✓ | X | ✓ | ✓ | X | ✓ | ✓ | X | ✓ |
| Manage co-existing problems/conditions | ✓ | X | ✓ | ✓ | ✓ | ✓ | ✓ | ✓ | – | – | – | – |
| Modify/understand physical environment | X | ✓ | ✓ | ✓ | ✓ | ✓ | ✓ | X | – | – | – | – |
| Enjoyment | X | ✓ | ✓ | X | X | ✓ | ✓ | ✓ | – | – | – | – |
| Modify/understand employment commitments | X | ✓ | X | X | X | ✓ | X | X | – | – | – | – |
| Increase/maintain daily activities | – | – | – | – | ✓ | ✓ | ✓ | X | – | – | – | – |

Lewthwaite et al. (2018), *PeerJ*, DOI 10.7717/peerj.4604

**Table 3** (*continued*)

| Theme | Physical activity | | | | Sedentary behaviour | | | | Sleep | | | |
|---|---|---|---|---|---|---|---|---|---|---|---|---|
| | COPD groups | | Expert groups | | COPD groups | | Expert groups | | COPD groups | | Expert groups | |
| | SA-COPD | NL-COPD | COPD-E | Non COPD-E | SA-COPD | NL-COPD | COPD-E | Non COPD-E | SA-COPD | NL-COPD | COPD-E | Non COPD-E |
| Understand cause/treat sleep problem | – | – | – | – | – | – | – | – | ✓ | ✓ | X | ✓ |
| Manage anxiety/stress/-worry | – | – | – | – | – | – | – | – | X | ✓ | ✓ | ✓ |
| Follow sleep hygiene principles | – | – | – | – | – | – | – | – | ✓ | ✓ | ✓ | ✓ |

**Notes.**

✓, important with high group agreement; X, theme not important with high group agreement; –, theme did not occur.

[a]Non-specific theme included items that were not able to be sorted into a common theme.

(**2**) people with COPD and experts largely agreed on what was important to improve behaviours; and (**3**) there was a disproportionate focus of items concerning increasing PA, particularly moderate-to-vigorous intensity PA (MVPA).

### What was important to people with COPD and experts?

A clear focus of important items and themes was disease management (important items: PA $n = 14$, 26%; SB $n = 7$, 29%; sleep $n = 1$, 33%). This included management of symptoms (breathlessness, fatigue)—the single theme important to all four participant groups to improve PA, SB and sleep—and management of co-existing conditions (cardiac conditions, pain, and anxiety). Disease management strategies important to our Delphi participants included optimisation of pharmacotherapy and professional support in the form of regular follow-up appointments with healthcare providers and/or supervised exercise programs (e.g., pulmonary rehabilitation). Each of these strategies are proven for the overall management of COPD (*Yang et al., 2017*; http://goldcopd.org/) and likely help to maintain active behaviours and good sleep quality by preventing acute COPD exacerbations (*Pitta et al., 2006*; *Gimeno-Santos et al., 2014*), alleviating symptoms (*Gimeno-Santos et al., 2014*; *Tödt et al., 2015*), reducing comorbid disease burden (*Gimeno-Santos et al., 2014*; *Miravitlles, Cantoni & Naberan, 2014*; *Sievi et al., 2015*), and improving exercise capacity (*Gimeno-Santos et al., 2014*). There is however limited evidence that optimising function with for example bronchodilator therapy (*Gimeno-Santos et al., 2014*; *Mantoani et al., 2016*) or pulmonary rehabilitation alone (*Cindy Ng et al., 2012*; *Soler, Diaz-Piedra & Ries, 2013*; *McDonnell et al., 2014*; *Geiger-Brown et al., 2015*; *Lahham, McDonald & Holland, 2016*; *Mesquita et al., 2017a*; *Mesquita et al., 2017b*) translates into long-term changes in habitual behaviour.

Important items and themes also commonly concerned the need to target behavioural factors (important items: PA $n = 33$, 61%; SB $n = 16$, 67%; sleep $n = 1$, 33%). To improve PA and SB, this included provision of physician advice or encouragement and intervention strategies relating to health counselling or self-management. Despite COPD clinical practice guidelines recommending physician advice and encouragement as a strategy to improve PA and SB (*Lewthwaite et al., 2017*), this has not been well explored. Counselling added to pulmonary rehabilitation has shown some promise for improving PA in people with COPD (*Lahham, McDonald & Holland, 2016*; *Mantoani et al., 2016*). However, there is currently no universally accepted approach for health counselling with interventions commonly comprised of various combinations of different behaviour change techniques (*Wilson et al., 2015*; *Williams et al., 2017*). As such, it is not clear which exact intervention components have led to the often small, positive effects on PA. The most common behaviour change techniques explored to date have included a combination of: social support, self-monitoring of behaviour, goal-setting/review of goals and information on when, where and how to perform the behaviour (*Wilson et al., 2015*; *Williams et al., 2017*). This overlaps with strategies important to our Delphi participants, which were self-monitoring, goal-setting, positive feedback and education on how and why to change behaviour. A number of items important to participants' concerned skill development for managing and/or coping with COPD (e.g., feel in control/empowered, self-regulate activity, persevere, follow action

plan). Typically, such factors would be addressed with self-management interventions (*Zwerink et al., 2014*; *Effing et al., 2016*) which to date, have not shown consistent positive effects on PA, SB or sleep (*Zwerink et al., 2014*). To improve sleep, education (on sleep hygiene principles) was the single important behavioural strategy. The effects of sleep hygiene education on sleep quality has not been well explored in people without a clinical sleep disorder (*Irish et al., 2015*), however it is commonly included in multicomponent behavioural interventions, which have shown small, positive effects on sleep quality and quantity in older adults (*Montgomery & Dennis, 2003*).

Few items and themes considered as important concerned the need to adapt the social or physical environments (important items: PA $n = 7$, 13%; SB $n = 1$, 4%; sleep $n = 1$, 33%). Where specific items were considered important to our Delphi participants, these concerned adapting the social environment (e.g., 'to have supportive family members …', 'to have social interactions with friends and families'). These findings overlap with two recent studies in the COPD population, where having an active resident loved one (*Mesquita et al., 2017a*; *Mesquita et al., 2017b*) and participating in active grand-parenting (*Arbillaga-Etxarri et al., 2017*) were associated with increased time spent in MVPA.

### Where did participant groups differ?

There were significant differences in importance rating of specific items within themes predominantly between the NL-COPD group and remaining groups. This may reflect some cultural differences. For example, while maintaining some level of employment was important to both COPD patient groups to improve PA and SB; volunteer work was rated significantly higher by Dutch participants. This is consistent with differences in volunteer participation between these two countries; an estimated 30% of retirees in Australia undertake volunteer work (*ABS, 2010*) compared to 50% in the Netherlands (*GHK, 2010*; *Cloïn, 2012*). This may reflect the greater promotion and support provided for volunteering at the local government level in the Netherlands (*Mensink, Boele & Van Houwelingen, 2013*).

Differences at the theme, rather than item level, were mostly seen with sleep. Compared to the remaining groups, few themes were important to the COPD expert group to improve sleep. It is possible that how we asked our Delphi participants about what is important for people with COPD to improve sleep may have resulted in some level of confusion around whether to treat sleep in this population as a medical problem or as a behaviour subject to change through general, non-clinical behaviour change strategies.

### Physical activity to improve all behaviours across the energy expenditure spectrum

A striking finding from this study was the clear focus on PA, particularly MVPA, by all participant groups. Participating in moderate-to-vigorous activities such as gym exercise, walking or cycling was frequently suggested by our Delphi participants as a way to improve all behaviours. This suggests that there exist implicit assumptions that:

a) MVPA is the primary movement-related behaviour that contributes to health;

b) Increasing MVPA is the same as reducing SB.

Perceptions around movement-related behaviours are likely to have been influenced by several decades of public health messaging promoting PA. While original activity guidelines developed in 1975 focused on higher intensity aerobic exercise for fitness (*Blair, LaMonte & Nichaman, 2004*), the focus has since evolved to consider a 24-h approach for health (*Chaput et al., 2014*; *Tremblay et al., 2016*). This approach takes into consideration the effect that all behaviours have on health: light-, moderate-, and vigorous-PA, SB, and sleep. The composition of these behaviours impacts health independently of MVPA (*Biswas et al., 2015*) and may enhance or attenuate the benefits achieved from participating in MVPA (*Chastin et al., 2015*). For the greatest health benefits, it may be advantageous to spend more time throughout the waking day in light activities than in SB or to substitute prolonged sleep for SB (*Chastin et al., 2015*). For sleep, optimal durations for adults have been shown to be between seven and nine hours (*Cappuccio et al., 2011*).

Importantly, SB is not just the obverse of MVPA. A relatively large quantitative change in MVPA (say increasing MVPA from 30 to 60 min/d) would have very little impact on SB (typically 8–10 h/d) even if a straight substitution occurred. To achieve meaningful reductions in SB it would be more feasible to modify or replace SB with light activities. This would require a different intervention approach to that for increasing MVPA (*Prince et al., 2014*).

## Strengths and limitations

This research study was strengthened by the systematic Delphi approach used to obtain participant perspectives. Following recommendations by *Diamond et al. (2014)* and *Sinha, Smyth & Williamson (2011)* for the Delphi process, criteria were set *a priori* for participant eligibility, number of questionnaire rounds, items to be removed from subsequent rounds and items to be considered as important. Furthermore, the Delphi process maintains participant anonymity avoiding common problems such as influence of a dominant group member or pressures to conform to group opinions. The research team were however ultimately responsible for a number of decisions that may have influenced items and themes considered as important. To minimise bias, a consensus approach was used where possible with two independent researchers and a third researcher available to resolve discrepancies where needed.

This study had a number of limitations. Participants for COPD groups were identified by pulmonary rehabilitation staff, with details on lung function or medical history not available to the researchers. As such, participant groups were described using self-report measures, obtained from the Round 1 questionnaire. The addition of objective data (e.g., spirometry and accelerometry) would permit participants to be classified according to disease severity (http://goldcopd.org/) and activity status to evaluate whether perceptions differ accordingly. In addition, while the four-round Delphi approach enabled items to be rated over consecutive rounds with use of controlled feedback, this may have induced participant fatigue, contributing to the loss of participants over subsequent questionnaire rounds particularly evidenced with the COPD-E group.

### How might the findings from this Delphi study inform future intervention research and clinical practice?

For all Delphi participants, there was clear consensus that symptom palliation (including breathlessness, fatigue, pain and anxiety) was perceived as a key mechanism for people with COPD to be more physically active, reduce time spent sitting and lying and improve sleep quality. Symptom palliation by necessity requires an understanding of each patient's concerns/fears/expectations, their symptom burden and their multi-morbidity in order to optimise medical management and structure appropriate multi-dimensional professional support systems (e.g., provision of follow-ups, pulmonary rehabilitation). When introducing specific behaviour change strategies, irrespective of the intervention type (e.g., self-management, behaviour counselling, etc.), starting with an explicit conversation that: (1) acknowledges the presence of problematic symptoms, and (2) provides clear explanation around how intervention goals relate to symptom palliation, may provide an effective avenue to further tailor and optimise these interventions to achieve enduring behaviour change. Behaviour change strategies important to our Delphi participants that could be tailored toward the goal of symptom palliation include: self-monitoring, goal setting, education on how and why to change behaviours, and positive feedback.

## CONCLUSION

This study was the first to obtain the perspective of people with COPD from different geographical locations and experts from COPD- and non-COPD-specific backgrounds on factors considered as important to improve PA, SB and sleep. Our Delphi participants perceived a multifactorial approach to be important which is consistent with previous research, though there was a disproportionate focus on increasing MVPA, with little consideration around optimising behaviours over the remainder of the day. There is need for future research and health-care providers to consider a 24-h approach for health, taking into consideration the entire spectrum of activities that make up a person's day and targeting strategies identified as important to people with COPD and experts accordingly.

### Funding

Hayley Lewthwaite is supported by an Australian Government Research Training Program Scholarship. The authors received no further financial support for the research, authorship, and/or publication of this article. The funders had no role in study design, data collection and analysis, decision to publish, or preparation of the manuscript.

### Grant Disclosures

The following grant information was disclosed by the authors:
Australian Government Research Training Program Scholarship.

### Competing Interests

The authors declare there are no competing interests.

## Author Contributions

- Hayley Lewthwaite conceived and designed the experiments, performed the experiments, analyzed the data, contributed reagents/materials/analysis tools, prepared figures and/or tables, authored or reviewed drafts of the paper, approved the final draft.
- Tanja W. Effing conceived and designed the experiments, analyzed the data, contributed reagents/materials/analysis tools, prepared figures and/or tables, authored or reviewed drafts of the paper, approved the final draft, provided translations for Dutch commincation/data.
- Anke Lenferink analyzed the data, contributed reagents/materials/analysis tools, prepared figures and/or tables, authored or reviewed drafts of the paper, approved the final draft, provided translations for Dutch communication/data.
- Tim Olds and Marie T. Williams conceived and designed the experiments, analyzed the data, contributed reagents/materials/analysis tools, prepared figures and/or tables, authored or reviewed drafts of the paper, approved the final draft.

## Human Ethics

The following information was supplied relating to ethical approvals (i.e., approving body and any reference numbers):

Ethical approval was granted by The Human Research Ethics Committees of Southern Adelaide Local Health Network (#516.15), Medisch Spectrum Twente (#K16-09) and University of South Australia (#0000034584).

## Data Availability

The raw data are provided as Supplemental File.

## Supplemental Information

Supplemental information for this article can be found online at http://dx.doi.org/10.7717/peerj.4604#supplemental-information.

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
