# Peer review of "Improving physical activity, sedentary behaviour and sleep in COPD: perspectives of people with COPD and experts via a Delphi approach"

_PeerJ, doi:10.7717/peerj.4604_

## Round 0.1 · original submission · Major Revisions

Thank you for your submission. It seems that including more information regarding the Delphi process would be of benefit to those unfamiliar with that process. Under procedure, you mention that participant anonymity was maintained. Please explain how. Under discussion, you mention that there are four key findings, but only list 3 of them. What is the fourth key finding? It would be helpful to include in the discussion any limitations of the study, you seem to focus only on the strengths in that section.

Reviewer 1 ·

Basic reporting

The quality of the writing in this manuscript was generally very good. However, several important details of the study were not well described (explained in detail below), making it difficult to assess/understand the methodology.

The introduction gave good context for the study, and cited references were relevant to the topic. Data was provided as supplemental files.

Experimental design

This manuscript falls within the scope of the journal. The research question is well defined and meaningful. As noted above, some lack of information/lack of clarity make it difficult to assess the methodology as currently written.

The authors have provided an ethic approval statement. This study appears to conform to the Declaration of Helsinki, although I could not find a copy of the consent form in the confidential supplemental materials. I did not see any identifiable information in the files. There were no unnecessary or unethical aspects to this study.

Validity of the findings

The above comments notwithstanding, the data appear robust. Conclusions are well stated and linked to the research question and current findings.

Additional comments

This paper is on an important and interesting topic. The research question makes intuitive sense, and is well supported by the introduction.

However, some important aspects of the methodology were not clearly described, making it difficult to evaluate the quality of the data.

1. It would be really helpful to clarify the Delphi process in more detail, for those of us unfamiliar with it. For example, I am unclear on the differences between rounds 2 and 3. Were items removed? Or were participants simply alerted to items that had low agreement? Were participants prompted to reconsider/change their response if there was low agreement? Were any items removed prior to round 4? If not, I'm unclear on the rationale for showing the list a third time. A thorough description of this process would be extremely helpful to the reader. Some of this information may be in Figure 1, but I found it very difficult to interpret without accompanying explanation in the text.
2. In Round 4 were participants providing agreement with the themes, rather than the individual items?
3. Line 177: The results jump from Round 1 to Round 4, with no discussion of Rounds 2 or 3. This may make sense within the Delphi process, but there should be some explanation of the reason for this for the reader.
4. Lines 139-143. I'm unclear on what is being described here. It sounds as though the results from round 1 are being compared to the results of round 1, and also to a systematic review? Were items from the review then added to the list as well?
5. What was the rationale for including experts from non-COPD specific backgrounds? I initially assumed that knowledge around COPD was what defined them as experts. A bit of background on the value of including these non-COPD experts would be very helpful, as the reason for their inclusion isn't explained/justified in the current introduction.
6. Related to the above, how was the number of COPD and non-COPD experts determined? It seems that the non-COPD expert category would be extremely large given how they were defined (e.g. likely 100’s, if not 1000’s of individuals worldwide). How was this winnowed down to a manageable number?
7. Line 100: More detail is needed on the types of websites and documents that were reviewed to identify experts. Was there specific criteria? Were search terms used? Or was this more of a convenience sample?
8. In the results section, the authors do not describe the items/themes that were rated with high agreement. Although this info is provided in tables, it is still worth summarizing in the text.
9. Lines 212-229: This is all good information, but it comes before any in-depth description of the results of this study itself, which is absent from the Results section. This information would be much more useful after a more thorough description of the specific findings of the Delphi process - e.g. what specific items/themes were important/unimportant, beyond MVPA.
10. Lines 47-48: these are good points, but the references are presented in such a way that it is unclear which refer to the general population, and which to people with COPD
11. Line 64: this sentence is worded a bit awkwardly. Are you saying that people with COPD view fatigue as having a bigger impact on QOL, and feel that daily activities are not affected by the disease? I suggest rewording, or perhaps breaking into 2 sentences, to make the meaning more clear.
12. Lines 73-74: The phrase " two countries where differences in factors to influence behaviours exist " is unclear to me. Do you mean the social and built environment differ between the two countries?
13. Did the authors consider looking at the views of different GOLD stages? It seems like stage I and Stage 4 could have very different views on what is important.
14. Were the questionnaires/results translated into Dutch by the authors? Did they have anyone back-translate the questionnaires to confirm they were consistent in both languages?
15. Line 200: was there disproportionate focus on increasing PA, or disproportionate agreement? Since the results aren't summarized in the text, it’s hard to know exactly what is being referred to.

Reviewer 2 ·

Basic reporting

No comment

Experimental design

- In the methodology (line 97), the authors state that patients were included if they had at least mild COPD as confirmed by spirometry. Do the authors have these spirometry data available? Were both COPD groups matched? (e.g. in terms of FEV1) Do the patients cover a wide spectrum of COPD patients in terms of obstruction severity? Is there information available on the number of acute exacerbations patients experienced in the past 12 months? These might have implication on the generalization of findings and is not reported in the table.

- How did the authors assess the number of days patients were active in the previous week? Was this evaluated with a questionnaire or objective data monitoring. Please specify in the methodology section. In case the authors used a questionnaire to analyze the level of physical activity, this should be mentioned in the 'limitations'-section.

- Line 147: After round 2, low group agreement was defined as [IQR] ≥2. On Line 151, low group agreement in round 4 was defined as IQR>2. Is there any rationale for this difference in cut-off value?

Validity of the findings

No comment

Additional comments

In this manuscript, Lewthwaite and colleagues identified a wide range of factors that are judged as important to improve behaviour (physical activity, sedentary behaviour and sleep) by patients with COPD (both from Australia and the Netherlands), COPD-experts and non-COPD specific experts via a Delphi approach. The investigators conducted a very well established and properly reported qualitative part. However, there is missing information to characterize both COPD groups more in detail (in terms of severity of airflow obstruction, past history of acute exacerbations, objective physical activity data,…) Addition of a section ‘clinical importance’ might be of great (additional) value to the manuscript in which the authors might summarize which specific themes they would include in the implementation of future interventions and how these themes should be tailored for each patient group.

Minor comments:

Table 1:
- Did the authors investigate any significant differences within each patient and expert groups? This was not reported but can play an important role in explaining differences in themes within each group.

Fig 2:
The percentages of the COPD-E and the Non-COPD-E groups are not clear. In the COPD-E group, 76 experts were invited of which 17 were include (22%). All those who were included participated to round 1A (which is still indicated as 22%). In round 1B, only 14 experts participated. However, this is indicated as 82%. This is not in line with the use of percentages in both patients group: Eighty patients from Australia were invited from which 29 were included (36%). Of these patients, 26 participated to round 1A (90%). If the same strategy would be applied as in the experts groups, the % would be 32.5 for round 1A in the Australian patients.

- Line 211: ‘Increasing MVPA is the same as reducing SB.’ The authors might consider to add the reference of Biswas et al., Ann Intern Med 2015. In their manuscript, they provide further evidence for the independent effect of sedentary time on health outcomes (independent from physical activity).

- Line 261: The authors refer to several behavioral change techniques. Can the authors further elaborate on the possible overlap of their findings with behavioral change techniques proposed in literature?

Reviewer 3 ·

Basic reporting

No comment

Experimental design

no Comment

Validity of the findings

no comment

Additional comments

This Delphi study is researching an important clinical area identifying the perspective of individuals with COPD and experts on what is considered important to improve behaviour and identifying areas of difference between these groups. Being an international collaborative study strengthens the findings and the overall, the manuscript is very well written, clear in the steps of the Delphi process and draws sounds conclusions based on the results found. Within the discussion, each of the key points have been well discussed in relation to existing literature, with the inclusion of factors unique to each country involved in this study, highlighting a global difference in strategies which may be applicable in future studies. The authors have highlighted the main strengths and limitations.

A few minor points for clarity are detailed below.
Abstract
Unclear what less commonly before point 3 means? Can the authors clarify this point?
Methods: a clear outline of the methods has been provided. Each step in the Delphi process is clearly articulated, with appropriate process of collation of themes by multiple pairs of authors.
Minor point
Page 11, line 122 – participant’s should read participants’
Line 132 – prior should read priori

---

## Round 0.2 · accepted · Accept

Thank you for your updated version of your manuscript. It appears you have addressed the reviewers concerns.

Reviewer 1 ·

Basic reporting

No comment

Experimental design

No Comment

Validity of the findings

No Comment

Additional comments

All of my previous comments were thoroughly addressed.

Reviewer 3 ·

Basic reporting

No comment

Experimental design

no comment

Validity of the findings

The authors have addressed all concerns

Additional comments

The authors have addressed all concerns well.